# Focused Ultrasound Combined with Microbubbles in Central Nervous System Applications

**DOI:** 10.3390/pharmaceutics13071084

**Published:** 2021-07-15

**Authors:** Ko-Ting Chen, Kuo-Chen Wei, Hao-Li Liu

**Affiliations:** 1Department of Neurosurgery, Linkou Chang Gung Memorial Hospital, Guishan, Taoyuan 333, Taiwan; chenkoting@gmail.com; 2Ph.D. Program in Biomedical Engineering, Chang Gung University, Guishan, Taoyuan 333, Taiwan; 3Neuroscience Research Center, Linkou Chang Gung Memorial Hospital, Guishan, Taoyuan 333, Taiwan; 4Department of Neurosurgery, New Taipei Municipal TuCheng Hospital, Chang Gung Medical Foundation, TuCheng, New Taipei 236, Taiwan; 5School of Medicine, Chang Gung University, Guishan, Taoyuan 333, Taiwan; 6Department of Electrical Engineering, National Taiwan University, Da’an, Taipei 106, Taiwan; 7Department of Biomedical Engineering, National Taiwan University, Da’an, Taipei 106, Taiwan

**Keywords:** focused ultrasound, central nervous system, microbubbles, blood–brain barrier

## Abstract

The blood–brain barrier (BBB) protects the central nervous system (CNS) from invasive pathogens and maintains the homeostasis of the brain. Penetrating the BBB has been a major challenge in the delivery of therapeutic agents for treating CNS diseases. Through a physical acoustic cavitation effect, focused ultrasound (FUS) combined with microbubbles achieves the local detachment of tight junctions of capillary endothelial cells without inducing neuronal damage. The bioavailability of therapeutic agents is increased only in the area targeted by FUS energy. FUS with circulating microbubbles is currently the only method for inducing precise, transient, reversible, and noninvasive BBB opening (BBBO). Over the past decade, FUS-induced BBBO (FUS-BBBO) has been preclinically confirmed to not only enhance the penetration of therapeutic agents in the CNS, but also modulate focal immunity and neuronal activity. Several recent clinical human trials have demonstrated both the feasibility and potential advantages of using FUS-BBBO in diseased patients. The promising results support adding FUS-BBBO as a multimodal therapeutic strategy in modern CNS disease management. This review article explores this technology by describing its physical mechanisms and the preclinical findings, including biological effects, therapeutic concepts, and translational design of human medical devices, and summarizes completed and ongoing clinical trials.

## 1. The Blood–Brain Barrier

The blood–brain barrier (BBB) is physically composed of tight junctions of the endothelial cells (EC) of capillary, pericyte, and astrocytic endfeet, and chemically comprises transporters that actively efflux materials away from the central nervous system (CNS) environment. The physical and chemical components of the BBB together form a dynamic, adaptable interface that maintains the homeostasis of the brain [1]. BBB dysfunction has been shown to be central to several CNS diseases, including multiple sclerosis, epilepsy, and stroke [2], whereas in other diseases such as Alzheimer’s disease (AD), the role of BBB disruption remains unclear [3]. Apart from primary dysfunction of the BBB in diseases, another aspect influencing therapeutic considerations is that 100% of molecules larger than 500 Da and more than 98% of all smaller molecules cannot cross the BBB [4]. This “barrier” aspect of the BBB significantly affects the ability of therapeutic agents to penetrate into the CNS and remains one of the main obstacles to treating diseases of the CNS, especially primary brain tumors and metastases.

In CNS malignancies such as gliomas [5] and metastases [6], the tumor cells not only disrupt the BBB while forming the blood–tumor barrier (BTB), but also form synapses with astrocytes and hijack CNS pathways to control the neurovascular unit in order to support their metabolic demands and tumor growth [4]. Despite the BTB being leakier than the BBB, it still hampers drug delivery due to its heterogeneous permeability and the presence of efflux transporters [7]. The heterogeneous permeability of the BTB in malignant brain tumor results from nonhomogeneous infiltration of tumor cells and unsynchronized neovascularization driven by the vascular endothelial growth factor. The portion of tumor margin with intact BBB leads to a marked reduced accumulation of therapeutic agents (decreasing of the enhanced permeability and retention effect). Studies have demonstrated primary brain tumors with clinically significant regions of intact BBB that warrant therapeutic targeting [8,9]. Such an intact BBB in the tumor microenvironment (TME) also plays a critical role in tumor chemoresistance by offering a hypoxic and acidic environment that is favorable to cancer stem cells [10,11]. In summary, the BBB, the BTB, and the intimately related microenvironment pose significant challenges and hence remain major obstacles to conquer in successfully treating brain diseases.

## 2. BBB Opening by Focused Ultrasound

### 2.1. Biological Effect Discovery

Vykhodtseva et al. first demonstrated that ultrasound exposure could affect the integrity of the BBB. A wide range of burst-type ultrasound parameters was tested to evaluate the biological effects of such ultrasound stimulation on the brain and to observe the correlation with the strength of wideband ultrasound. Ultrasound at a high intensity of 7000 W/cm^2^ induced histological effects including BBB disruption, hemorrhage, tissue necrosis, and animal death, whilst a dose-dependent relationship between injury severity and ultrasound parameters was identified [12]. Hynynen et al. were the first to apply burst-type focused ultrasound (FUS) sonication in the presence of intravenously administered microbubbles; they observed histological effects on the brain [13]. The addition of circulating microbubbles allows BBB opening (BBBO) to be achieved at an ultrasound intensity at least three orders of magnitude lower than when microbubbles are absent. The intensity threshold for inducing BBBO can be lower than the threshold for brain damage, implying safety in future applications [13]. The reversibility of FUS-induced BBBO (FUS-BBBO) has also been confirmed by the using the contrast agent gadolinium-diethylenetriamine pentaacetic acid (Gd-DTPA) in contrast-enhanced magnetic resonance imaging (CE-MRI). When using a 690 kHz probe, CE-MRI demonstrated that 50%, 90%, and >90% of sonicated areas were enhanced (indicating BBBO) at pressures of 0.4, 0.8, and 1.4 MPa, respectively. The T1-weighted images showed dose-dependent contrast enhancement and normalization of the signal intensity (SI) change at 5 h after sonication, suggesting the utility of CE-MRI with Gd-DTPA as a noninvasive biomarker for evaluating the dynamics of BBBO [14].

### 2.2. Histological Findings and Tissue Damage

There were early attempts to test the histological effects over a wide range of ultrasound intensities on FUS-BBBO. Hynynen et al. reported using 1.5 MHz ultrasound at pressures from 2 to 12.7 MPa and pulse repetition frequencies up to 1000 Hz in 10 µs bursts [15]. They used an exposure intensity that was about two orders of magnitude higher than current typical parameter settings and observed vascular wall damage, tissue necrosis, ischemia, and apoptosis at a pressures of 6.3 MPa. Reducing the acoustic pressure significantly reduced red blood cell (RBC) extravasation and avoided the occurrence of ischemia, tissue necrosis, and apoptosis, showing the dependence of tissue damage on FUS intensity [15]. McDannold et al. examined a lower acoustic exposure range to test whether delayed effects exist for FUS-BBBO [16]. Under an appropriate microbubble dose and longer bursts (100 ms), and pressures from 0.7 to 1 MPa at an ultrasound frequency of 1.63 MHz, they found small-scale RBC extravasation, with minimal apoptosis and ischemia. Moreover, no histological abnormalities was observed 4 weeks after sonication, implying the absence of a delayed effect of BBBO [16].

In addition to histological examinations, T2*-weighted MRI and magnetic resonance (MR) susceptibility-weighted imaging have been reported to be suitable for detecting hemorrhage accompanying FUS-BBBO. Liu et al. confirmed the occurrence of microhemorrhage during BBBO for excessive exposure intensities [17]. In addition, B-mode ultrasound imaging can be utilized to reveal blood clots as hyperechoic signals, whereas microbubble ultrasound contrast-enhanced imaging can be utilized to detect transient blood flow imbalances and shortages due to the presence of intracranial hemorrhage [17,18].

Arvanitis et al. used a lower exposure intensity that was near the inertial cavitation threshold as indicated by wideband emission in passive cavitation detection (PCD). With 5 min of sonication, tissue ischemic necrosis was observed in the targeted regions, which coincided with the wideband emission signature and hence confirmed the occurrence of inertial cavitation. This approach further reduced the power required to induce therapeutic ablation in the brain by nearly twofold, and reduced the possibility of thermal damage adjacent to sensitive structures near the skull base (e.g., optic tract) [19].

### 2.3. Inflammatory Effect

FUS-BBBO is a mechanical process that disrupts tight junctions, which not only induces imbalance in vascular–extravascular homeostasis, but also potentially triggers an immunological response. Kovas et al. first demonstrated the triggering of a sterile inflammatory response (SIR) that was characterized by the expression of damage-associated molecules. Histological analysis confirmed the appearance of TUNEL^+^ neurons, infiltrated CD68^+^ macrophages, activated astrocytes and microglia, and the enhanced expression of cell adhesion molecules in the vasculature. Through transcriptomic analysis they identified that the SIR stemmed from regulation of the NFκB pathway [20].

McMahon et al. used microarray analysis of the hippocampal microvasculature to demonstrate the increased transcription of proinflammatory cytokine genes at 6 h after FUS-BBBO, which returned to baseline at 24 h after sonication [21]. They also investigated the effects of different doses of microbubbles, and found that the NFκB pathway was only activated at higher microbubble doses, suggesting that inducing BBBO using clinically accepted microbubble doses may effectively reduce the potential hazards of brain edema, neurodegeneration, neutrophil inflammation, and microhemorrhage [22].

Sinharay et al. employed the positron-emission tomography (PET) tracer [^18^F]-DPA714 to assess the neuroinflammatory response and found that this was strongly correlated with the activation of microglias and astrocytes accompanying BBBO. Moreover, the inflammatory response was visualized only at 24 h postsonication and did not accumulate after repeated FUS stimulation, implying that multiple sonications could be utilized for future clinical applications [23]. In addition, Zhao et al. proposed using a microbubble–nanoparticle complex system produced by conjugating phosphatidylserine to specifically trigger microglia activation and minimize the potential hazard caused by the widespread activation of microglia [24].

Among these groups, Zhao’s group focused more on the inflammatory response by inducing cerebral ischemic reperfusion injury in a model but not the neuroinflammatory effect induced by FUS-BBBO [24]. Among the other four studies, three of them used the same device with various concentration of microbubbles [21,22,23] while one used different device and higher microbubble concentration (100 μL of Optison) [20]. Two clear messages have been delivered from their results: (1) FUS-BBBO induced acute inflammatory response is dependent on microbubble dose [20,22]; and (2) a PCD feedback control provides real-time monitoring for safe BBBO and prevents significant inflammatory damage even after repeated sonications [20,21,22,23].

### 2.4. Safety of Repeated Interventions

Single sessions of FUS-BBBO have been confirmed to be feasible and safe, and several studies have also investigated the effect of repeated FUS-BBBO. McDannold et al. found no behavioral deficits, visual defect, or loss in visual acuity after repeated FUS-BBBO in the central visual field over a period of several weeks in primates who were trained to conduct complex visual acuity tasks [25]. In addition, Downs et al. investigated the safety of repeated BBBO in the caudate and putamen in primates over 4–20 months. CE-MRI confirmed the absence of tissue edema and hemorrhage; however, reaction times in cognitive, motivational, motor, and behavioral tasks were temporarily prolonged, although they returned to baseline within 5 days [26].

Using an implanted device, Horodyckid et al. verified the safety of long-term repeated FUS-BBBO in primates over 4 months. They found no change in cerebral glucose metabolism in ^18^F-fluorodeoxyglucose (FDG) PET images, no epileptic signals, and no abnormal nerve conduction in electroencephalography and somatosensory evoked potential (SSEP) recordings [27]. Tsai et al. conducted frequent BBBO two or three times weekly for several weeks in a mouse model. Ultrasound combined with excessive microbubble doses (0.4 mL/kg) was also compared with the standard dose (0.15 mL/kg) [28]. The results demonstrated that frequent BBBO with excessive exposure produced minor and short-term behavioral changes, while frequent BBBO with typical exposure did not cause behavioral or histological changes. It was concluded that safety concerns with frequent and repeated FUS-BBBO can be minimized and controlled by using a standard microbubble dose [28].

### 2.5. Vascular Observations

Cho et al. and Nhan et al. used two-photon fluorescence microscopy to detect the status of the capillary bed and microvasculature during FUS-BBBO. They observed the leaking dynamics of injected dextran. Faster dextran accumulation (which occurred instantaneously with BBB disruption) was found to be accompanied with absolute and immediate BBB disruption, whereas slower dextran accumulation (5–15 min postsonication) occurred over a wide range of ultrasound intensities [29,30]. Tsai et al. recently used swept-scan optical coherence tomography and real-time 3D angiography to reveal transient cerebral vessel dilation during FUS-BBBO, where the degree of vessel dilation was correlated with the FUS intensity [31].

## 3. BBBO Optimization

### 3.1. Medical Image Detection

CE-MRI was the first in vivo tool available for BBBO detection, since the molecular mass of ~938 Da of Gd-DTPA means that this molecule cannot penetrate into the parenchyma with focal BBBO [13]. Other than Gd-DTPA, the use of superparamagnetic iron-oxide (SPIO) nanoparticles that were previously used as a blood-pool contrast agent in T2-weighted imaging has been investigated for detecting BBBO. The larger dimensions of SPIO (33–90.6 nm) impede its ability to cross the BBB, and therefore the spin–spin (R2) relaxation cannot provide sufficient sensitivity. Liu et al. demonstrated the use of T2*-weighted imaging to amplify the magnetic field susceptibility, which then provided sufficient R2 level changes that were correlated with the degree of BBBO, making it feasible for detecting FUS-BBBO [32]. One unique advantage of using T2*-weighted imaging with SPIO is that these magnetic nanoparticles can be actively guided using external magnets to improve the targeting of their deposition. This characteristic allows active SPIO targeting after BBBO, and has been shown to further enhance the delivery of a conjugated therapeutic agent with magnetic SPIO nanoparticles compared with using FUS-BBBO alone [33]. In addition, Wang et al. used gold nanorods (AuNR) that can be excited by a tunable laser and the BBBO can be monitored by photoacoustic imaging. Long-term photoacoustic imaging after FUS-BBBO revealed local parenchyma retention of AuNR owing to its lengthy retention period preventing it from metabolically clearing from the extravascular space [34].

To understand the pharmacodynamics of BBBO, Vlachos et al. employed dynamic CE-MRI (DCE-MRI) to characterize the permeability change by measuring the accumulation of Gd-DTPA in the extravascular–extracellular space (EES) [35]. Park et al. subsequently used the transfer coefficient (Ktrans) of Gd-DTPA to correlate BBBO with drug concentration (doxorubicin) during FUS-BBBO [36]. A linear correlation was found between Ktrans and the doxorubicin concentration, implying that kinetic DCE-MRI can be used to estimate local drug concentrations. Chai et al. analyzed the two-directional permeability dynamics during FUS-BBBO. In addition to Ktrans, which describes the outflow permeability from the capillary toward the EES, the counterpermeability (Kep), which describes the permeability from the EES toward capillaries, was also monitored. For ultrasound pressures ranging from 0.4 to 0.8 MPa, the measured half-life values of Ktrans were 2.8–5.3 h, whereas those of Kep were significantly longer, at 17–90 h. Such imbalance in the Ktrans/Kep ratio reveals the possibility of enhanced drug retention in the EES for the purpose of drug delivery [37]. This result also implies the existence of active P-glycoprotein pumping activation during the BBB restoration process, which has been confirmed histologically by Aryal et al [38].

In addition to DCE-MRI, Lin et al. used the single-photon-emission computed tomography (SPECT) tracer ^99m^Tc-DTPA (molecular mass: ~487 Da) to detect BBBO. They showed that the radiotracer activity saturated at 1.5 h post sonication, indicating the hemodynamics of the ^99m^Tc-DTPA exchanged from the intravascular to the extravascular space. The radioactive counts were also correlated with the degree of BBBO, demonstrating the possibility of using SPECT to detect BBBO [39]. Figure 1 summarizes representative images for the various modalities used to detect and monitor BBBO.

### 3.2. Microbubbles and Ultrasound Parameters

Choi et al. investigated the effects of the microbubble size on the efficacy of BBBO. Microbubbles were synthesized with a DSPC (1,2-disearoyl-snglycero-3-phosphocholine) and PEG40S (polyoxyethylene-40 stearate) lipid shell and a PFB (perfluorobutane) core [46], and microbubbles with diameters in three ranges of 1–2, 4–5, and 6–8 µm were isolated. A bolus of 1 mL/kg at a concentration of 8.5 × 10^8^ microbubbles/mL was injected. Compared with microbubbles with a diameter of 6–8 µm, those with a diameter of 1–2 µm required a higher acoustic pressure to induce the same degree of BBBO, whereas microbubbles with a diameter of 4–5 µm could result in a similar degree of BBBO at a lower pressure [47]. Samiotaki et al. investigated how the microbubble size was related to the BBB recovery for opening periods ranging from 24 h to 5 days. The BBB recovery was associated with the microbubble size, with larger microbubbles inducing a higher degree of BBBO as well as resulting in a longer recovery time [46].

Yang et al. used commercially available phospholipid-coated microbubbles (SonoVue) with a mean diameter of 2.5 µm (diameter range of 1–5 µm) at concentrations of 1–5 × 10^8^ microbubbles/mL. Three different doses were applied (0.15, 0.3, and 0.45 mL/kg), which were somewhat higher than the SonoVue doses of 0.05–0.1 mL/kg used in diagnostic clinical applications. The use of a higher concentration induced a higher degree of BBBO [48]. McDannold et al. compared FUS-BBBO between two commercially available microbubbles: Optison (mean diameter 1.1–3.3 µm) and Definity (mean diameter 2–4.5 µm); the former is albumin-shelled and the latter is lipid-shelled, and both are widely distributed polydisperse microbubbles. Although sonication in the presence of Optison produced with slightly higher degree of BBBO and a tendency for more RBC extravasation, the probability of BBBO was virtually the same [49]. Wu et al. compared the efficacy of FUS-BBBO among three commercially available microbubbles, SonoVue, Definity, and USphere (mean sizes of 1 µm), and found that the degree of BBBO was similar for all three types at the same microbubble concentration (4 × 10^7^ microbubbles/mL) [50]. McMahon et al. compared Definity with two specially designed microbubbles: BG8774 (polydisperse, with a mean diameter similar to that of Definity) and MSB4 (monodisperse, with a mean diameter of 4 µm). The degree of BBBO induced with BG8774 was similar to that induced with Definity at the same concentration (20 µL/kg). In contrast, MSB4 induced a significantly lower degree of BBBO, which was mainly due to the shear stress induced by the monodisperse microbubbles (concentration equivalent to 20 μL (liquid volume)/kg of Definity) insufficient to disrupt tight junctions [51].

With a bolus type of injection, all microbubbles experience a short circulation time and the effect of BBBO is significantly degraded when the interval between injections exceeds 4 min. A compensatory strategy for inducing uniformly distributed BBBO using multiple sonications by adjusting the exposure time for each sonication (i.e., the earliest sonication is associated with the shortest sonication time) has been proposed [50]. McDannold et al. investigated how ultrasound parameters influence BBBO. A longer burst length induced a significantly higher degree of BBBO was found. The FUS pressure thresholds for BBB disruption were estimated to be 0.69, 0.47, and 0.36 MPa for burst durations of 0.1, 1, and 10 ms, respectively [49].

### 3.3. Detection and Control

In order to characterize the correlation between BBBO and acoustic sonication, McDannold et al. conducted in vivo experiments with passive cavitation detection (PCD) to simultaneously record the backscattered acoustic emissions [52]. They showed that RBC extravasation was associated with wideband emissions, while BBBO can be induced in the absence of wideband emissions, suggesting that tissue damage would not necessarily accompany successful BBBO. In addition, the second and third harmonics were strongly associated with BBBO [52]. On the other hand, Tung et al. utilized spectrograms to reveal that the pressure threshold for BBBO (0.3 MPa) was lower than that for inertial cavitation (0.45 MPa), indicating that BBBO occurred before inertial cavitation took place [53]. The signature of the harmonics therefore has the potential to serve as an indicator for controlling BBBO. Using a MR-compatible FUS system, Arvanitis et al. found that increases in the harmonics of the acoustic emissions were strongly correlated with the accumulation of Gd-DTPA in CE-MRI. The increases in the harmonics were then used as an indicator to predict successful BBBO, with a high predictive rate of 96% [54].

O’Reilly et al. employed PCD to capture and characterize the ultraharmonic components in real time from the emissions. The FUS intensity was gradually increased until ultraharmonics were identified, and then the FUS intensity was reduced. For a reduction of up to 50%, a sufficient BBBO could be achieved with minimal tissue damage [55]. Similar to the controlled scheme proposed above, Huang et al. used a 3 × 3 scanning grid to create a large-volume BBBO in a trans-skull large-animal (porcine) model. Their results suggested the feasibility of using this scheme to control BBBO across animals of different sizes [56]. Furthermore, Sun et al. proposed the use of a closed-loop cavitation control algorithm based on the idea of sustaining stable cavitation while suppressing inertial cavitation [57]. The method was used to deliver a chemotherapeutic agent (liposomal doxorubicin) in a mouse glioma model. The addition of the closed-loop control improved the delivery of liposomal doxorubicin compared with the uncontrolled group, while also minimizing potential tissue damage.

A special dual-frequency confocal transducer has been employed to perform PCD-based BBBO control. Tsai et al. demonstrated a simple implementation of using a confocal concentric dual-frequency transducer (550/1100 kHz) for transmitting 1100 kHz ultrasound and receiving acoustic emissions at 550 kHz [58]. Since 550 kHz piezoceramics exhibit a narrow bandwidth, it was observed that the acoustic emissions were strongly correlated with BBBO. One unique advantage of this arrangement is its simplicity to colocalize two transducers to capture subharmonic emissions originating from the targeted position to improve the spatial-discrimination capability. Employing such a configuration for real-time control has been reported to exhibit excellent sensitivity (92%) and specificity (92.3%) in detecting BBBO [58].

## 4. Preclinical Validation of CNS Disease Treatment

### 4.1. BBBO for Brain Tumors (Smaller Drug Molecules)

FUS-BBBO can be used to enhance the administration of anticancer drugs used to treat brain tumors, including temozolomide (TMZ; molecular mass: 194 Da), carmustine (214 Da), carboplatin (371 Da), irinotecan (587 Da), doxorubicin (543 Da), and paclitaxel (853 Da) [40,59,60,61,62,63,64]. For drugs that can already penetrate the BBB, such as TMZ, which appears in the cerebrospinal fluid (CSF) and plasma at a ratio of 20%, we previously showed that the addition of BBBO can increase the TMZ concentration almost twofold, to a CSF/plasma ratio of 38% [60,61]. BBBO also lengthens the half-life of TMZ inside the brain parenchyma. The combination of BBBO with TMZ therefore seems to show a prominent synergistic therapeutic effect [60,61]. For drugs that cannot normally cross the BBB, Treat et al. first demonstrated that BBBO facilitated a large dose of doxorubicin penetrating into the brain, up to 886–5336 ng/g, consistent with the therapeutic concentration of >700 ng/g [62]. In addition, a FUS-BBBO approach increased the concentration of carboplatin by 7.3-fold in normal tissue and 2.9-fold in a tumor (glioma). Moreover, the tumor doubling time was prolonged by 96% compared with carboplatin-alone animals, which resulted in a 48% increase in the median survival [59]. Similarly, without BBBO, the concentration of irinotecan in the brain has been reported to be very low, with a tissue/plasma ratio of 2%, whereas after BBBO the concentration ratio increased to 178% [63]. FUS-BBBO also increased the concentration of paclitaxel by 3–5 times [64]. Liposomal doxorubicin is another form of doxorubicin that utilizes liposome as a drug carrier to prolong its half-life in plasma [57,65,66]. Several preclinical studies found that FUS-BBBO significantly increased the concentration of liposomal doxorubicin 5–10-fold compared with the control group, to reach the therapeutic dose in the brain (up to 4800 ng/g) [65,66].

### 4.2. BBBO for Brain Tumors (Large Molecular Drugs)

The delivery of large molecular drugs across the BBB has also been demonstrated, including IL-12 (70 kDa) [67], dopamine D-4 receptor-targeting antibody [68], and HER2/c-erbB2 (humanized antihuman epidermal growth factor receptor 2) monoclonal antibody [69]. We have demonstrated the enhanced delivery of bevacizumab (149 kDa) in a mouse glioma model [45]. Bevacizumab is an antiangiogenic monoclonal antibody that inhibits tumor growth by interfering with neovascularization in the TME. However, its anticancer effect is subsequently hampered by “vascular normalization” via the pruning effect induced by itself. The addition of FUS-BBBO increased drug penetration by up to fivefold and significantly increased the median survival by up to 135% compared with the non-BBBO group [45].

In addition to glioma, whether BBBO benefits brain metastasis treatment has also been investigated. Using a brain metastasis animal model mimicking HER2^+^ breast cancer, FUS-BBBO plus trastuzumab significantly prolonged survival compared with trastuzumab alone. It is noteworthy that a portion of the animals in the BBBO-plus-trastuzumab group showed complete tumor regression, indicating the potential therapeutic value of this approach [70].

### 4.3. BBBO Anticancer Immune Modulation

Excessive FUS-BBBO accompanied with microhemorrhage can result in SPIO-labeled macrophages being temporarily recruited and accumulating at the BBBO site, implying that FUS-BBBO exposure may trigger immune cell homing effect [70]. Liu et al. identified the recruitment of tumor-infiltrating lymphocytes (TILs) including CD8^+^ cytotoxic T lymphocytes (CTLs) and CD4^+^ non-Treg (regulatory T cells) TILs locally at the targeted tumors, with unchanged levels of CD4^+^CD25^+^ cells. It is noteworthy that the CD8^+^/CD25^+^ ratio increased, indicating enhancement of the anticancer activity, which supported the potential anticancer response [70,71,72]. In a subsequent study, Chen et al. redesigned the experiments in a brain tumor xenograft model. FUS-BBBO plus an intraperitoneally administered immune triggering agent (IL-12) resulted in a profound increase in all TILs, including the CD8^+^ CTLs, CD4^+^ T helper cells, and Tregs, as well as in the CD8^+^/CD25^+^ ratio. Such a prominent immune response associated with IL-12 has been demonstrated to benefit tumor control and survival [67,73].

More recently, Chen et al. reported mouse glioma experiments accompanying a neuronavigation-guided FUS-BBBO clinical trial for recurrent glioblastoma (rGBM) patients aimed at examining whether an immunostimulatory response was triggered 7 days after FUS exposure. For a clinically safe BBBO parameter (0.63 mechanical index (MI)), the immune cell population did not change noticeably; however, at a higher but still safe dosage (0.81 MI, which caused minimal RBC extravasation), noticeable increases in CD4^+^ and CD8^+^ TILs were observed. These observations support the possibility of fine tuning the FUS parameters to induce an appropriate anticancer immune response while simultaneously maintaining the treatment safety [74].

In addition to tumor models, an immunomodulatory effect such as the activation of innate immune cells (microglia) has been shown to play an important role in beta amyloid clearance to benefit AD treatment in animal experiments [75,76]. The activation of the proinflammatory pathway triggered by microbubble–ultrasound interactions was demonstrated. However, for the tumor microenvironment (TME), it remains unknown whether the immune response triggered by FUS-BBBO benefits brain tumor treatment.

### 4.4. BBBO for Alzheimer’s Disease Treatment

Raymond et al. successfully used FUS-BBBO to deliver both therapeutic and molecular imaging agents in AD mouse models [77]. Jordao et al. subsequently demonstrated that the targeted delivery of BAM-10 anti-beta-amyloid antibodies rapidly reduced plaque pathology in an AD mouse model [78]. Burgess et al. produced similar findings when targeting the hippocampus. The behavioral improvement correlated with the reduction of amyloid plaques, with accompanying evidence of hippocampal neurogenesis [75]. Leinenga et al. elegantly demonstrated that 75% of scanning-ultrasound-treated animals exhibited reductions in the plaque burden via microglial activation. The reduction of plaques resulted in improvements in the memory and behavioral performance of the animals [76].

Tau pathology is another attractive therapeutic target in both AD and other tauopathies. FUS-BBBO combined with RN2N tau-specific antibodies has been demonstrated to focally reduce the phosphorylation of tau protein and relieve anxiety-like behavior [79]. This blockade of the phosphorylation pathway was also confirmed in Hsu et al. [80], in which they delivered enhanced GSK-3 (glycogen-synthase kinase-3) inhibitor into an AD animal model and found a reduction of amyloid plaques as well as an effective reduction in GSK-3 activity of up to 61.3%.

IVIg (intravenous immunoglobulin) is a fractionated human blood product containing polyclonal antibodies that acts as an immunomodulator both peripherally and centrally. With the help of FUS-BBBO, it has recently been reported that BBBO with the enhanced IVIg deposition downregulated proinflammatory cytokine TNF (tumor necrosis factor)-alpha to modulate inflammatory effects, boost neurogenesis, and reduce amyloid plaques in hippocampus evaluated in a transgenic AD animal model [81]. Since the impairment of neurogenesis and accumulation of amyloid plaques are both pathognomonic of AD, FUS-BBBO opens the potential to increase the efficacy of immunotherapy for AD treatment.

### 4.5. BBBO for Parkinson’s Disease Treatment

FUS-BBBO can also aid the delivery of therapeutic genomic materials including gene-encoded vectors, RNA interference (RNAi), and proteins (e.g., neurotrophic factors) to treat CNS diseases such as Parkinson’s disease (PD) and Huntington’s disease. Baseri et al. and Wang et al. demonstrated the feasibility of delivering neurotrophic factors including brain-derived neurotrophic factor (BDNF), glia-derived neurotrophic factor (GDNF), and neurturin. The bioactivity of these factors was found to be preserved following the conjugation, administration, and delivery processes [82,83]. 

In addition to directly delivering therapeutic proteins into the brain, another strategy is to enhance the delivery of viral vectors that can express such proteins in the CNS. Hsu et al. demonstrated that FUS-BBBO successfully delivered the recombinant adeno-associated virus (AAV)-2 reporter gene into the brain. The expression peaked at 2 weeks after FUS exposure, and the expression level was strongly correlated with the CE-MRI SI immediately after ultrasound exposure, indicating that reporter gene expression can be predicted by CE-MRI [43]. Noroozian et al. reported that FUS-BBBO reduced the titer of the intravenous administration of recombinant AAV by 100-fold while producing equivalent AAV expression in the brain [84].

Designing a nonviral vector by conjugating DNA plasmid on liposomes is another option for carrying genes. One advantage of using a nonviral gene vector is the reduced concern about immunogenicity during systemic administration. Lin et al. demonstrated that using gene-loaded liposomes (reporter gene plasmid conjugated with the lipid material to form liposomes) significantly increased the expression efficiency (by fivefold) in FUS-BBBO compared with direct injection [85]. Moreover, in contrast to expression peaking at 2 weeks when using a viral vector, the expression of this liposome-based nonviral vector peaked after 2–3 days [85]. One challenge when using a nonviral vector for gene delivery is that the CNS expression is inferior to that with a viral vector. It is possible to enhance the expression by synthesizing lipid-based microbubbles with directly conjugating the plasmid to form gene-vector microbubbles or a liposomal microbubble complex. Microbubble oscillations will expose the vector adjacent to the tight-junction pores so as to experience the largest shear stresses, which increases the probability of the vector genes penetrating and hence improves gene transfection. Lin et al. attempted to conjugate gene-carrying plasmid liposome with microbubbles to form a gene-liposome–microbubble complex that would enhance the delivery of the GDNF gene [86]. In the MPTP animal model mimicking PD, the successful delivery of the GDNF gene was confirmed with the expression of reporter gene. The gene-liposome–microbubble complex has shown superior performance in gene delivery efficiency than using gene liposome and microbubbles separately during FUS exposure [86].

Fan et al. further improved the lipid material by forming cationic microbubbles, which have a higher affinity to gene material. The cationic plasmid–microbubble complex has been found to provide a higher gene payload and improve the reporter gene/GDNF expression in the brain after FUS-BBBO. In the 6-OHDA model mimicking PD, the cationic plasmid–microbubble complex outperformed the neutral form in terms of gene expression efficiency, and superior animal behavioral recovery was observed [87]. Long et al. used a similar approach by conjugating gene plasmid with microbubbles. They demonstrated that the therapeutic Nrf2 (nuclear factor E2-related factor 2) gene can be overly expressed in the targeted brain region and revealed a possible mechanism for neuroprotection for PD treatment [88].

There have been recent further attempts to concurrently carry GDNF and BDNF genes in the gene-liposome–microbubble complex. Parallel evaluations were made of the neuroprotective effects when (1) delivering GDNF, (2) delivering BDNF, and (3) combining delivery of GDNF and BDNF by examining the pathological changes in dopaminergic neurons [89]. The results demonstrated that both BDNF and GDNF gene delivery via FUS-BBBO provided a sufficiently high neuroprotective effect with evidence of behavioral improvement; decreased calcium influx, GFAP, and caspase 3 expression; and rescued dopaminergic neuronal loss. However, concurrently delivering GDNF and BDNF genes did not provide additional benefits, possibly due to competing expression between these genes [89].

### 4.6. BBBO for Huntington’s Disease Treatment

Huntington’s disease is an inherited disease that affects the central nervous system and causes progressive degeneration of brain cells and leads to deterioration of motor skills and cognitive abilities as well as behavioral difficulties. Burgess et al. attempted to utilize FUS-BBBO to enhance the delivery of RNAi with the intention of specifically decreasing the expression of the mutant Huntingtin protein (Htt). It has been shown that successfully delivering small interfering RNA directly to the striatum significantly reduced the expression of Htt. In addition, the reduction of Htt expression at the FUS treatment site was strongly correlated with the degree of BBBO [90]. In addition, Lin et al. described the strategy by using BBBO to enhance the delivery of the GDNF-plasmid–liposome complex in a mouse model. The enhanced GDNF expression resulted in significantly decreased formation of polyglutamine-expanded aggregates, reduced oxidative stress and apoptosis, promoted neurite outgrowth, improved neuronal survival, and improved motor performance of mice [91].

Table 1 summarizes preclinical experimental results of using different mechanisms provided by FUS-BBBO in treating various disease animal models.

## 5. Clinical Translation of BBBO

### 5.1. Medical Device Design

FUS-BBBO has been demonstrated preclinically to allow the delivery of various therapeutic agents into the CNS, prolong survival in CNS malignancies, and facilitate recovery in neurodegenerative diseases, which has prompted various investigations into clinical applications in human subjects. Currently three different concepts of system design for human applications are being developed and undergoing clinical investigations. 

The first system design is adapted from the device originally approved for essential tremor via thermal ablation (Insightec ExAblate Neuro Type I; center frequency: 650 kHz) [92]. A modification was made to fit the purpose of BBBO (Insightec ExAblate Neuro Type II; center frequency: 220 kHz) [93]. More than 1000 elements of the hemispherical phased array allow electrically steering with a wide range around the geometrical center. Motion during the FUS intervention is prevented by using a four-pin stereotactic frame. Using frame-based stereotactic target selection and confirmation with intraoperative MRI allows the focus to be precisely targeted. The discomfort associated with stereotactic pinning was reported to be tolerable. MR thermometry is used to confirm that the temperature increase induced by the focal beam is within the safe range before applying the therapeutic excitation. The use of phased array means that the focus beam can be electrically steered to induce multiple targets according to the planned volume geometry [94,95]. The treatment takes 2–4 h. The strengths of the MR-guided FUS (MRgFUS) BBBO procedure are accurate guidance and targeting. However, the price paid is a loss of flexibility (needs to be operated in an MR suite) and high system design complexity due to the requirement of full MR compatibility. 

The second system design involves implanted ultrasound ceramics, where the transducer disc is placed into a bur hole made on the skull (SonoCloud, Carthera). Since the ultrasound energy can be transmitted without cranial-bone-induced sound blockage, the use of a planar ultrasound disc has been confirmed to deliver sufficient ultrasound energy for inducing BBBO. The device is implanted intraoperatively, oftentimes after tumor resection and before wound closure, and ultrasound conduction can be achieved by simply reconnecting the signal wiring using a punching procedure on the scalp flap [96]. The strength of the system is its simplicity in conducting ultrasound-induced BBBO. However, a major inconvenience is that the implantation of the transducer disc needs to be determined based on the predicted disease progression. A new version of the system involves implanting nine emitters while creating multiple bur holes so that a wider area can be covered to vary or widen the BBBO region for better disease control [97]. 

The third system employs a neuronavigation system to guide BBBO. Neuronavigation was originally designed for guiding surgical tools, whereby a registration process using patient anatomical imaging, MRI, or computed tomography is used to interactively demonstrate the position of registered tools on a 3D anatomical image with high accuracy. A guiding procedure is adopted to guide the aiming point of the invisible focal beam via a simple coordinate transformation or an aiming axis extension designated from the console interface of the system. The error in large-animal BBBO was reported to be ~2.3 mm, indicating that the guiding accuracy is acceptable [98]. An advantage of this system is its relatively low integration complexity, with it being feasible to incorporate the FUS apparatus with currently commercially available neuronavigation systems. This also means that the system could deliver treatments even in the outpatient setting, and a recent study found that the overall procedure can be accomplished within half an hour [74]. However, a major concern is that the guidance relies on perfect ballistic beam projection offered from numerical treatment planning without actual beam deposition. There are currently two devices available that are based on neuronavigation guidance: the first one utilizes a commercially available neuronavigation system whose first clinical trial results have been reported (NaviFUS, Taiwan) [74], and the other device from Columbia University incorporates the neuronavigation system into the FUS generator and is currently under clinical investigation (NCT04118764).

### 5.2. Clinical Brain Tumor Treatment

Carpentier et al. conducted the first BBBO clinical trial (NCT02253212) using an implanted planar ultrasound device (SonoCloud, Carthera) to enhance carboplatin delivery in rGBM patients. The interim results of the phase I/II dose-escalating trial in 15 patients demonstrated successful BBBO in the FUS beam path [96]. The final results showed that BBBO could be detected by CE-MRI in 52 (80%) of 65 BBBO sessions performed in 19 patients. The median progression-free survival and overall survival were 2.73 and 8.64 months in patients with poor or no BBBO, compared with 4.11 and 12.94 months in those with clear BBBO [99]. So far that is the only study to demonstrate a potential survival benefit using ultrasound-induced BBBO to enhance drug delivery in brain tumor patients.

Mainprize et al. reported the use of an MRgFUS system (ExAblate Neuro, Insightec) in a BBBO clinical trial (NCT02343991) for malignant glioma patients. The trial treated five patients (one using liposomal doxorubicin delivery and four using TMZ delivery) with BBBO immediately before their tumor debulking surgery, so that the drug concentrations in the sonicated samples could be measured. The biochemical analyses of paired peritumoral samples revealed that the drug concentration was higher at sonicated than unsonicated sites, demonstrating the potential of enhanced drug delivery [100]. Park et al. applied a multiple-BBBO trial (NCT03712293) on the same day of TMZ chemotherapy using MRgFUS to newly diagnosed rGBM patients. Five out of six patients completed the full six cycles of BBBO sessions accompanying the adjuvant TMZ treatments. BBBO was confirmed at a median of 94.3% of the targets by CE-MRI, and none of the patients experienced FUS-BBBO-related complications [101,102].

Chen et al. reported on the use of the NaviFUS neuronavigation-guided FUS system in a BBBO clinical trial (NCT03626896) in rGBM patients. This single-arm, feasibility, dose-escalated, nonrandomized trial recruited six rGBM patients and used FUS exposure intensities ranging from 0.48 to 0.68 MI. The primary endpoint of BBBO and patient safety were achieved. Moreover, a dose-dependent effect of FUS energy on BBBO was found on DCE-MRI, and no ultrasound-related adverse effect was reported [74,103].

The results of the aforementioned trials show the feasibility of all three devices in treating malignant brain tumor patients with high tolerability and few procedure-related side effects. Moreover, integrating BBBO drug delivery into standard care provided a potential survival benefit.

### 5.3. Clinical AD Treatment

Lipsman et al. first reported the use of MRgFUS for BBBO in AD patients. They recruited five patients with mild-to-moderate AD, and two FUS treatments were applied with an interval of 1 month targeting the right dorsolateral prefrontal cortex. No difference in beta amyloid burden on [^18^F]-florbetaben PET imaging or in cognitive scores were found at the 3-month evaluation [95]. A particularly interesting finding was that a functional connectivity analysis with resting-state functional MRI (fMRI) demonstrated a transient deterioration in the ipsilateral frontoparietal network, but with no long-term changes in the frontoparietal or default mode network 3 months later [104].

Based on the preclinical support of FUS-BBBO triggering neurogenesis in the hippocampus, a clinical validation to verify its benefit for AD treatment was conducted. Metha et al. reported the use of MRgFUS for BBBO in early AD patients and evaluated the fluid flow patterns using serial CE-MRI (NCT03671889). Three participants underwent multiple treatment sessions of BBBO targeting the hippocampus and entorhinal cortex. It was found that CE reappeared in the perivenular regions downstream from the targeted sites after BBB closure, supporting that the permeability of the blood–meningeal barrier may last longer, suggesting a perivenular immunological healing response [105].

In summary, FUS-BBBO has been employed clinically to treat mild-to-moderate AD. MRgFUS has been tested to confirm the safety of repeated BBBO in AD patients [94,95]. A significant temporary decrease in functional connectivity of the frontoparietal network was observed [104], supporting the previous reported neuromodulation effects accompanying BBBO [106,107,108].

### 5.4. Clinical Adoption for Other Diseases

Amyotrophic lateral sclerosis (ALS), a motor neuron degenerative disorder, has also been investigated clinically. Abarhao et al. employed MRgFUS in a single-arm trial (NCT03321487) and demonstrated safe and successful BBBO in the primary motor cortex (arm and leg areas as localized by task-related fMRI). No motor complication was detected [109]. Gasca-Salas et al. conducted a single-arm trial for PD dementia (PDD) (NCT03608553) using an MRgFUS system. Five PDD patients were recruited, and the BBBO target was the right parieto-occipito-temporal cortex. Two sessions of FUS were given with an interval of 2–3 weeks and without side effects. There were mild cognitive improvements, but no major changes in amyloid plaques were detected using ^18^F-FDG PET [110,111].

Table 2 summarizes the completed and ongoing FUS-BBBO trials related to brain tumors and other diseases.

## 6. Other CNS Applications

### 6.1. BBBO-Induced Neuromodulation and Sonogenetics

FUS-BBBO induces permeability changes in vascular endothelial structures, triggers the activation of microglia and astrocytes, and induces proinflammatory responses, but the associated mechanical stresses also directly induce biophysical effects on neurons (neuromodulation). McDannold et al. reported on the preclinical utilization of FUS-BBBO to enhance systemically delivered GABA in the primary somatosensory cortex, which then temporarily suppressed neural activity as well as the SSEP produced by electrical stimulation of the sciatica nerve [107]. Similarly, Chu et al. used fMRI blood-oxygen-level-dependent (BOLD) signals and SSEP recordings to test the effect of FUS alone on the somatosensory cortex. Neuronal activity was observed to be temporarily suppressed after FUS exposure in a dose-dependent manner. While exposure at 0.55 MI induced short SSEP transients after sonication for 60 min, exposure at 0.8 MI resulted in long-term suppression for 1 week. This SSEP suppression coincided with depression of the BOLD signals on fMRI, indicating true FUS-BBBO-induced neuromodulation [108]. Todd et al. further analyzed changes in functional connectivity between the sensory cortex hindlimb region and other cortices after FUS-BBBO using resting-state fMRI [106]. They found significant reductions in connectivity between targeted BBBO regions and other cortices, with no effect on the connectivity in untreated areas. Moreover, the alterations seemed to recover by 1 week after FUS-BBBO. That study clearly demonstrated a neuromodulatory effect of FUS-BBBO, and the underlying mechanism may have had both vascular and neuronal origins [106]. Cui et al. demonstrated that reducing the ultrasound energy in the presence of microbubbles so as to not induce BBBO could still produce neuronal excitation [112]. This observation indicates that neural stimulation can originate from ultrasound-induced amplification of the radiation force and microstreaming of microbubbles and might not require the occurrence of BBBO. 

Activating a selective group of neurons in a specific location requires even more precision and cell-type selectivity compared with modulating all neurons. Ibsen et al. demonstrated the combined use of microbubbles with low-intensity ultrasound to stimulate neurons transfected by the TRP-4 mechanosensitive ion-channel genes in *Caenorhabditis elegans*. The transfected neurons were sensitized to an ultrasound stimulus, which induced a behavior outcome, and hence the method has been called “sonogenetics” [113]. Another approach using genetically modified neurons containing the mechanosensitive transmembrane protein prestin (an auditory-sensing protein) has been demonstrated to detect an ultrasound stimulus. Once prestin was noninvasively transfected and expressed in the animal brain, ultrasound successfully triggered neuron activity, thereby indicating sonogenetic neuronal manipulation [114].

### 6.2. FUS-Mediated Thrombolysis

It has been reported that combining diagnostic ultrasound with microbubbles can accelerate the fibrinolysis induced by urokinase (54 kDa) [115] and recombinant tissue plasminogen activator (rtPA, 70 kDa) [116], supporting that acoustic cavitation is one mechanism for accelerating thrombolysis. Similarly, Lee et al. showed in an in vivo model that low-dose rtPA combined with FUS can produce thrombolysis equivalent to that for full-dose rtPA without FUS [117]. Wang et al. further developed a clot-targeted microbubble that induced a superior clot-lysis effect compared with the untargeted group using 800 kHz therapeutic ultrasound [118]. Ren et al. proposed another strategy of using a conjugated thrombolytic drug (urokinase) with microbubbles to synergistically induce the clot-lysis effect as well as to reduce the likelihood of urokinase-induced hemorrhage. In an in vitro setup, they demonstrated that the urokinase–microbubble complex provided better bioactivity than that in the microbubble-alone and urokinase-alone groups [119].

### 6.3. FUS-BBBO to Sensitize Liquid Biopsy

FUS-BBBO can actually work bidirectionally, with permeability from the EES to the vascular circulation allowing biomarker detection in the peripheral blood that would be blocked naturally by the BBB. Zhu et al. first demonstrated the concept of using FUS-BBBO as a noninvasive liquid biopsy tool. Intracranial injections of enhanced green fluorescent protein (eGFP)-transduced glioblastoma cells have been conducted in two animal models (U87 and GL261). The plasma eGFP mRNA level was significantly higher in animals with FUS-BBBO than in untreated animals [120]. The same group further reported increases in the concentrations of CNS proteins, glial fibrillary acidic protein, and myelin basic protein after FUS sonication compared with before treatment in a large-animal (porcine) model [121].

For clinical verifications, Meng et al. compared the plasma samples obtained from two clinical trials: NCT03618680 and NCT03739905. Their results demonstrated that FUS-BBBO increased the plasma concentrations of cell-free DNA, neuron-derived extracellular vesicles, and brain-specific protein S100b. This demonstrated the potential of implementing FUS-BBBO for liquid biopsies in the clinic [122].

### 6.4. Opening the Brain–Retina and Brain–Spinal Cord Barriers 

Similar to the BBB, the blood–retina barrier (BRB) and the blood–spinal cord barrier (BSCB) are composed of tight junctions of the endothelium and impede drug delivery due to the presence of efflux transporters. It is therefore reasonable to assume that concepts of FUS-BBBO can be adopted for opening these other barriers. Park et al. applied FUS exposure to the retina via the cornea and lens and assessed the BRB opening (BRBO) effect via Gd-DTPA leakage in CE-MRI. BRBO can be achieved at pressures similar to those used for BBBO (0.8 to 1.1 MPa), with gradual closing subsequently occurring within 3 h. RBC extravasation and scattered petechiae were observed only occasionally [123].

For the BSCB, Weber-Adrian et al. presented the use of FUS-induced BSCB opening (FUS-BSCBO) with the aim of delivering self-complementary adeno-associated virus serotype 9 (scAAV9) carrying reporter gene GFP. The results demonstrated that FUS-BSCBO delivered scAAV9 into the spinal cord and successfully expressed GFP for a major population of neurons [124]. Payne et al. verified the feasibility of FUS-BSCBO in both small and large animals. In a porcine model, a specialized dual-aperture configuration of the FUS setup was employed to maximize the acoustic window when stimulating the spinal cord with the focal beam. At a nonderated pressure of up to 4 MPa, ultrasound combined with microbubbles confirmed the feasibility of efficiently permeating the BSCB (in 16 of 24 cases), while histology examinations confirmed minimal tissue damage. This confirms that FUS-BSCBO has potential in clinical practice [125].

## 7. Concluding Remarks and Future Perspectives

It is 100 years since the term “blood–brain barrier” was first used by Stern [126], since when the sanctity of the brain has been attributed to tight regulation of its environment by both physical and chemical barriers. The BBB is also troublesome once pathogens or neoplasms reside within the CNS sanctuary due to it excluding the vast majority of therapeutic options. Numerous methods have been proposed for overcoming the BBB, but they have had major drawbacks, including imprecise temporal and spatial resolution of BBBO and the invasiveness of the methods [127]. Here we have demonstrated that combining FUS with the systemic administration of microbubbles significantly pushes the therapeutic boundary of CNS disease by precisely, transiently, reversibly, and noninvasively inducing BBBO.

There has been significant progress in therapeutic ultrasound over the past 2 decades, which provides hope for CNS diseases that were previously undertreated due to hinderance by the BBB. Malignant brain tumor is at the top of the list due to its worst survival, followed by neurodegenerative diseases with a lack of efficient options for reversing deterioration processes. However, certain problems still need to be overcome before adopting this approach in clinical settings. Firstly, a larger volume of FUS-BBBO is needed to provide sufficient impact regarding the infiltrative nature of malignant brain tumors or global decline in cognitive dysfunction associated with degenerative diseases. Secondly, several mechanisms are induced by FUS-BBBO, including drug delivery, immunomodulation, vascular modification and oxygenation, and neuromodulation, and various FUS parameters need to be tailored to maximize the therapeutic effects and minimize the adverse effects in different diseases. Finally, as a new modality of therapeutic tools, modification and optimization are needed for FUS-BBBO to allow its incorporation into standard care, and properly designed clinical trials are mandatory to demonstrate its clinical benefits.

## Figures and Tables

**Figure 1 pharmaceutics-13-01084-f001:**
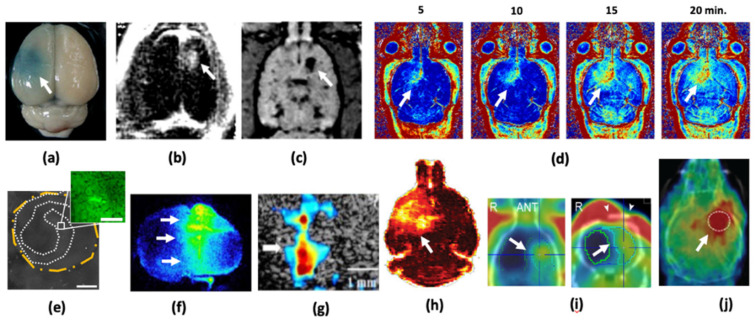
Representative images demonstrating the use of various modalities to monitor FUS-BBBO: (**a**) staining with Evans blue dye (arrow) [40], (**b**) TI-weighted CE-MRI [33], (**c**) T2-weighted CE-MRI [41], (**d**) MRI T1 relaxometry to sequentially follow the distribution of penetrating Gd-DTPA [42], (**e**) fluorescent-tag dextran penetration using optical microscopy [43], (**f**) autoradiography [39], (**g**) dynamic ultrasound imaging [44], (**h**) DCE-MRI [37], (**i**) ^99m^Tc-DTPA SPECT [39], and (**j**) ^68^Ga PET [45] (min. = minutes, ANT = anterior; Adapted from Chen et al., Frontiers Media S.A., 2019).

**Table 1 pharmaceutics-13-01084-t001:** Summary of preclinical experimental results of different mechanisms of FUS-BBBO in various disease models.

Disease Model	Pathogenesis and Unmet Need of Disease	Therapeutic Effect Induced by FUS-BBBO	Agents of Delivery or Reaction Related to FUS Treatment		Main Results
			Category	Therapeutic Agents	
Primary brain tumor—glioblastoma	Infiltrative growth of glioma cells limiting surgical total removal nearly impossible	Enhance drug delivery	Smaller molecules	Temozolomide [60,61], carmustine [40], carboplatin [59], irinotecan [63], doxorubicin [62], paclitaxel [64], liposomal-doxorubicin [57,65,66]	Increase drug concentration in all studies, and potential survival benefit [45,59]
			Larger molecules (>1 kDa)	IL-12 [67], dopamine D4 receptor-targeting antibody [68], humanized antihuman EGFR2 monoclonal antibody [69], bevacizumab [45]	
		Enhance anticancer immunity		Macrophage [70], TILs [67,73,74]	
Metastatic brain tumor	Small and multiple metastasis, refractory to systemic therapy	Enhance drug delivery		Tratuzumab [71]	Cause tumor regression and survival benefit [71]
Alzheimer’s disease	1. Progressive formation and accumulation of amyloid plaques and tau proteins 2. Degeneration of hippocampal neurons	Clear amyloid plaques		Antiabeta antibody (BAM) [78], RN2N tau specific antibody [79], GSK-3 inhibitor, IVIG [81]	Decrease amyloid plaques, boost neurogenesis, improve behavior performance
		Enhance immunity		Microglia [75,76]	
		Neurogenesis		Hippocampal neurogenesis [75]	
Parkinson’s disease	Progressive degeneration of dopaminergic motor neurons in substantia nigra	Neurogenesis	Gene-encoded viral vector	Recombinant AAV-2 [43,84]	Enhance gene expression and enhance neurotrophic factors delivery at targeted region, even behavioral improvement [87,89]
			Gene-encoded nonviral vector	Liposomal plasmid [85], gene-liposome–microbubble complex [86,89], cationic plasmid microbubble [87], Nrf2 gene plasmid–microbubble [88]	
			Neurotrophic factors	BDNF, GDNF, neuturin [82,83]	
Huntingtin disease	Genetically inherited mutant Htt overproduction to damage neurons	Increase the expression of the mutant Htt		RNAi [90]	Reduce Htt expression

AAV: adeno-associated virus; BDNF: brain-derived neurotrophic factor; GDNF: glia-derived neurotrophic factor; Htt: Huntingtin protein; IVIG: intravenous immunoglobulin; RNAi: RNA interference.

**Table 2 pharmaceutics-13-01084-t002:** Summary of completed and ongoing clinical trials of FUS-BBBO.

Trial No.	Study Title	Indication	Microbubble/Drug	Device/ Treatment Cycle/ Parameters	Location	Status	Main Results
Brain Tumors							
NCT02253212	Safety of BBBO with SonoCloud	rGBM (*n* = 27)	SonoVue (0.1 mL/kg)/carboplatin	SonoCloud/multiple/0.5–1.1 MPa	France	Completed [96]	Repeated BBBO in combination with carboplatin was safe.
NCT03626896	Safety of BBB disruption using NaviFUS system in rGBM multiforme patients	rGBM (*n* = 9)	SonoVue (0.1 mL/kg)	NaviFUS/single/escalated exposure average 10–16 W	Taiwan	Completed [74]	Targeted and reversible BBBO was safely induced.
NCT03712293	ExAblate BBB disruption for glioblastoma in patients	Glioblastoma (*n* = 10)	Definity (4 μL/kg)/standard chemotherapy	ExAblate Neuro/multiple/PCD-based power regulation	Korea	Completed [101]	Multiple BBBO in combination with temozolomide was safe.
NCT03714243	BBB disruption using MRgFUS in the treatment of HER2^+^ breast cancer brain metastases	Breast cancer with brain metastases (*n* = 10)	Definity (4 μL/kg)/trastuzumab	ExAblate Neuro/multiple/PCD-based power regulation	Canada	Recruiting	Not available
NCT04446416	Efficacy and safety of NaviFUS system with add-on bevacizumab in rGBM patients	rGBM (*n* = 10)	SonoVue (0.1 mL/kg)/bevacizumab	NaviFUS/multiple/PCD-based power regulation	Taiwan	Recruiting	Not available
NCT03616860	Assessment of safety and feasibility of ExAblate BBB disruption for treatment of glioma	Glioblastoma (*n* = 20)	Definity (4 μL/kg)/TMZ	Insightec/multiple/PCD-based power regulation	Canada	Recruiting	Not available
AD							
NCT02986932	BBBO using FUS with intravenous contrast agents in patients with early AD	AD (*n* = 6)	Definity (4 μL/kg)	ExAblate Neuro/multiple/PCD-based power regulation (average 4.6 W)	Canada	Completed [94]	Targeted BBBO was safe and precise without inducing group-wise amyloid change.
NCT03119961	BBBO in AD	AD (*n* = 10)	SonoVue (0.1 mL/kg)	SonoCloud/multiple/0.5–1.1 MPa	France	Completed	Not available
NCT03671889	ExAblate BBB disruption for the treatment of AD	AD (*n* = 20)	Definity (4 μL/kg)	ExAblate Neuro/multiple/PCD-based power regulation	USA	Recruiting [95,104]	FUS-BBBO transiently affect frontoparietal network function.
NCT03739905	ExAblate BBBO for treatment of AD	AD (*n* = 30)	Definity (4 μL/kg)	ExAblate Neuro/multiple/PCD-based power regulation	Canada	Recruiting	Not available
NCT04118764	Noninvasive BBBO in AD patients using FUS	AD (*n* = 6)	Definity (10 μL/kg)	Single-element exploratory device/multiple	USA	Recruiting	Not available
PD and Others							
NCT03608553	Evaluate temporary BBB disruption in patients with PDD	PDD (*n* = 10)	Definity (4 μL/kg)	ExAblate Neuro/multiple/PCD-based power regulation	Spain	Not yet recruiting	Not available
NCT04250376	Use of transcranial FUS for the treatment of neurodegenerative dementias	PDD (*n* = 10)	Luminity (4 μL/kg)	ExAblate Neuro/multiple/PCD-based power regulation	USA	Recruiting [110,111]	Repeated BBBO is safe and may induce mild cognitive improvement.
NCT03321487	BBBO using MR-guided FUS in patients with ALS	ALS (*n* = 8)	Definity (4 μL/kg)	ExAblate Neuro/multiple/PCD-based power regulation	Canada	Recruiting [109]	FUS-BBBO in motor cortex was safe.

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
