# Peer review of "Focused Ultrasound Combined with Microbubbles in Central Nervous System Applications"

_pharmaceutics, 2021, doi:10.3390/pharmaceutics13071084_

Round 1

Reviewer 1 Report

The authors reviewed the progress of focused ultrasound (FUS) combined with microbubbles to induce the local detachment of tight junctions of capillary endothelial cells without inducing neuronal damage, precise, transient, reversible and noninvasive the blood-brain barrier opening (BBBO) in the treatment of some central nervous system (CNS) diseases. The pre-clinical and clinical trial are included as well as future perspectives. Overall, it is a good review paper of a very important and promising technology. After some modification it can be published.

The title is suggested to be focused on the focused ultrasound induced the blood-brain barrier.

Section of 6.2, FUS-mediated thrombolysis, may be removed to address only FUS-BBBO

FUS parameters, microbubble concentrations, and the other application specifications should be included when stating the bioeffects.

Line 100 specify “blood products as hyperechoic signals”

Line 137 specify “single FUS-BBBO”, single session of FUS-BBBO or FUS-BBBO only at one very small location?

Line 151 specify “small-animal model”

Line 152 specify “standard dose”

Line 185 What’s “the long”?

Line 228 change “a larger degree” to “a higher degree”, thereafter

Line 246 change “similar” to “similar to”

Line 249 What’s the concentration of “monodispersed microbubbles”

Line 261 change “in-vivo” to “in vivo” as well as “in-vitro” to “in vitro”, thereafter

Line 302 change the citation from [59],[40,60-64] to [40,59-64]

Table 1 it is a little hard to read this table, for example, what’s the relationship of “enhance drug deliver” and “enhance anti-cancer immunity” in the first row with the category listed in the same row?

Reviewer 2 Report

This article is a very comprehensive summary. The review article explores this novel and attractive approach of FUS-induced BBB opening to perform localized CNS therapeutic agent delivery, summarizes the existing preclinical findings and current ongoing clinical trials. The authors have read a lot of literature and grasped the contents accurately. Through reading this review, especially translational application from preclinical to clinical, we are full of expectations for the application of this technology.

However, some questions are still remaining and authors should address those questions.

  1. Figure 1 is similar to Figure 2 in the author's previous article (Chen KT, Wei KC, Liu HL. Theranostic Strategy of Focused Ultrasound Induced Blood-Brain Barrier Opening for CNS Disease Treatment. Front Pharmacol. 2019;10:86.). Is this appropriate?
  2. Is it possible to add main results like those in Table 1 to Table 2? These can help to understand the conclusions of clinical trials.
  3. Can ultrasonic parameters be added in Table 1, which is conducive to comparing the differences of ultrasonic parameters in each experiment?
  4. Can the ultrasonic intensity should be added in various clinical trial of Table 2?

Reviewer 3 Report

Chen et al. present a review on the use of focused ultrasound combined with microbubbles in central nervous system. The authors have covered the main applications of this technology.

  • Specific comments :

Paragraph 1. Line 49-51. It would be interesting to give more details about the BTB heterogeneous permeability.

Paragraph 2.2 line 83-85. In the previous paragraph 2.1, some studies presented are using US only to open BBB. Could you precise if microbubbles were used in the cited study (15).

Paragraph 2.3 Inflammatory effect. Could you precise the impact of US acoustic pressures on inflammatory effects. I presume the cited studies in this paragraph are not using the same parameters, could you precise this ?

Figure 1. Did you check if you have the rights to use the pictures ?

Table 1.

The table is hard to read, lines should be added to subdivide; for example between small and large molecules.

line Parkinson disease. Correct the following typo “gene encoded non vital vector”.

Round 2

Reviewer 3 Report

The authors answered correctly all the comments.